# Practical Methods for Graph Two-Sample Testing

**Debarghya Ghoshdastidar**
Department of Computer Science
University of Tübingen
ghoshdas@informatik.uni-tuebingen.de

**Ulrike von Luxburg**
Department of Computer Science
University of Tübingen
Max Planck Institute for Intelligent Systems
luxburg@informatik.uni-tuebingen.de

## Abstract

Hypothesis testing for graphs has been an important tool in applied research fields for more than two decades, and still remains a challenging problem as one often needs to draw inference from few replicates of large graphs. Recent studies in statistics and learning theory have provided some theoretical insights about such high-dimensional graph testing problems, but the practicality of the developed theoretical methods remains an open question.

In this paper, we consider the problem of two-sample testing of large graphs. We demonstrate the practical merits and limitations of existing theoretical tests and their bootstrapped variants. We also propose two new tests based on asymptotic distributions. We show that these tests are computationally less expensive and, in some cases, more reliable than the existing methods.

## 1 Introduction

Hypothesis testing is one of the most commonly encountered statistical problems that naturally arises in nearly all scientific disciplines. With the widespread use of networks in bioinformatics, social sciences and other fields since the turn of the century, it was obvious that the hypothesis testing of graphs would soon become a key statistical tool in studies based on network analysis. The problem of testing for differences in networks arises quite naturally in various situations. For instance, Bassett et al. (2008) study the differences in anatomical brain networks of schizophrenic patients and healthy individuals, whereas Zhang et al. (2009) test for statistically significant topological changes in gene regulatory networks arising from two different treatments of breast cancer. As Clarke et al. (2008) and Hyduke et al. (2013) point out, the statistical challenge associated with network testing is the curse of dimensionality as one needs to test large graphs based on few independent samples. Ginestet et al. (2014) show that complications can also arise due to the widespread use of multiple testing principles that rely on performing independent tests for every edge.

Although network analysis has been a primary research topic in statistics and machine learning, theoretical developments related to testing random graphs have been rather limited until recent times. Property testing of graphs has been well studied in computer science (Goldreich et al., 1998), but probably the earliest instances of the theory of random graph testing are the works on community detection, which use hypothesis testing to detect if a network has planted communities or to determine the number of communities in a block model (Arias-Castro and Verzelen, 2014, Bickel and Sarkar, 2016, Lei, 2016). In the present work, we are interested in the more general and practically important problem of two-sample testing: *Given two populations of random graphs, decide whether both populations are generated from the same distribution or not.* While there have been machine learning approaches to quantify similarities between graphs for the purpose of classification, clustering etc. (Borgwardt et al., 2005, Shervashidze et al., 2011), the use of graph distances for the purpose of hypothesis testing is more recent (Ginestet et al., 2017). Most approaches for graph testing based on classical two-sample tests are applicable in the relatively low-dimensional setting, where the

population size (number of graphs) is larger than the size of the graphs (number of vertices). However, Hyduke et al. (2013) note that this scenario does not always apply because the number of samples could be potentially much smaller — for instance, one may need to test between two large regulatory networks (that is, population size is one). Such scenarios can be better tackled from a perspective of high-dimensional statistics as shown in Tang et al. (2016), Ghoshdastidar et al. (2017a), where the authors study two-sample testing for specific classes of random graphs with particular focus on the small population size.

In this work, we focus on the framework of the graph two-sample problem considered in Tang et al. (2016), Ginestet et al. (2017), Ghoshdastidar et al. (2017a), where all graphs are defined on a common set of vertices. Assume that the number of vertices in each graph is $n$, and the sample size of either population is $m$. One can consider the two-sample problem in three different regimes: **(i)** $m$ is large; **(ii)** $m > 1$, but much smaller than $n$; and **(iii)** $m = 1$. The first setting is the simplest one, and practical tests are known in this case (Gretton et al., 2012, Ginestet et al., 2017). However, there exist many application domains where already the availability of only a small population of graphs is a challenge, and large populations are completely out of bounds. The latter two cases of small $m > 1$ and $m = 1$ have been studied in Ghoshdastidar et al. (2017a) and Tang et al. (2016), where theoretical tests based on concentration inequalities have been developed and practical bootstrapped variants of the tests have been suggested. The contribution of the present work is three-fold:

1. For the cases of $m > 1$ and $m = 1$, we propose new tests that are based on asymptotic null distributions under certain model assumptions and we prove their statistical consistency (Sections 4 and 5 respectively). The proposed tests are devoid of bootstrapping, and hence, computationally faster than existing bootstrapped tests for small $m$. Detailed descriptions of the tests are provided in the supplementary material.

2. We compare the practical merits and limitations of existing tests with the proposed tests (Section 6 and supplementary). We show that the proposed tests are more powerful and reliable than existing methods in some situations.

3. Our aim is also to make the existing and proposed tests more accessible for applied research. We provide Matlab implementations of the tests in the supplementary material.

The present work is focused on the assumption that all networks are defined over the same set of vertices. This may seem restrictive in some application areas, but it is commonly encountered in other areas such as brain network analysis or molecular interaction networks, where vertices correspond to well-defined regions of the brain or protein structures. Few works study the case where graphs do not have vertex correspondences in context of clustering (Mukherjee et al., 2017) and testing (Ghoshdastidar et al., 2017b, Tang et al., 2017). But, theoretical guarantees are only known for specific choices of network functions (triangle counts or graph spectra), or under the assumption of an underlying embedding of the vertices.

**Notation.** We use the asymptotic notation $o_n(\cdot)$ and $\omega_n(\cdot)$, where the asymptotics are with respect to the number of vertices $n$. We say $x = o_n(y)$ and $y = \omega_n(x)$ when $\lim_{n\to\infty} \frac{x}{y} = 0$. We denote the matrix Frobenius norm by $\|\cdot\|_F$ and the spectral norm or largest singular value by $\|\cdot\|_2$.

## 2 Problem Statement

We consider the following framework of two-sample setting. Let $V$ be a set of $n$ vertices. Let $G_1, \ldots, G_m$ and $H_1, \ldots, H_m$ be two populations of undirected unweighted graphs defined on the common vertex set $V$, where each population consists of independent and identically distributed samples. The two-sample hypothesis testing problem is as follows:

*Test whether $(G_i)_{i=1,\ldots,m}$ and $(H_i)_{i=1,\ldots,m}$ are generated from the same random model or not.*

There exist a plethora of nonparametric tests that are provably consistent for $m \to \infty$. In particular, kernel based tests (Gretton et al., 2012) are known to be suitable for two-sample problems in large dimensions. These tests, in conjunction with graph kernels (Shervashidze et al., 2011, Kondor and Pan, 2016) or distances (Mukherjee et al., 2017), may be used to derive consistent procedures for testing between two large populations of graphs. Such principles are applicable even under a more general framework without vertex correspondence (see Gretton et al., 2012). However, given graphs

on a common vertex set, the most natural approach is to construct tests based on the graph adjacency matrix or the graph Laplacian (Ginestet et al., 2017). To be precise, one may view each undirected graph on $n$ vertices as a $\binom{n}{2}$-dimensional vector and use classical two-sample tests based on the $\chi^2$ or $T^2$ statistics (Anderson, 1984). Unfortunately, such tests require an estimate of the $\binom{n}{2} \times \binom{n}{2}$-dimensional sample covariance matrix, which cannot be accurately obtained from a moderate sample size. For instance, Ginestet et al. (2017) need regularisation of the covariance estimate even for moderate sized problems ($n = 40, m = 100$), and it is unknown whether such methods work for brain networks obtained from a single-lab experimental setup ($m < 20$). For $m \ll n$, it is indeed hard to prove consistency results under the general two-sample framework described above since the correlation among the edges can be arbitrary. Hence, we develop our theory for random graphs with independent edges. Tang et al. (2016) show that tests derived for such graphs are also useful in practice.

We assume that the graphs are generated from the inhomogeneous Erdős-Rényi (IER) model (Bollobas et al., 2007). This model has been considered in the work of Ghoshdastidar et al. (2017a) and subsumes other models studied in the context of graph testing such as dot product graphs (Tang et al., 2016) and stochastic block models (Lei, 2016). Given a symmetric matrix $P \in [0,1]^{n \times n}$ with zero diagonal, a graph $G$ is said to be an IER graph with population adjacency $P$, denoted as $G \sim \text{IER}(P)$, if its symmetric adjacency matrix $A_G \in \{0,1\}^{n \times n}$ satisfies:

$$(A_G)_{ij} \sim \text{Bernoulli}(P_{ij}) \text{ for all } i < j, \quad \text{and} \quad \{(A_G)_{ij} : i < j\} \text{ are mutually independent.}$$

For any $n$, we state the two-sample problem as follows. Let $P^{(n)}, Q^{(n)} \in [0,1]^{n \times n}$ be two symmetric matrices. Given $G_1, \ldots, G_m \sim_{\text{iid}} \text{IER}\left(P^{(n)}\right)$ and $H_1, \ldots, H_m \sim_{\text{iid}} \text{IER}\left(Q^{(n)}\right)$, test the hypotheses

$$\mathcal{H}_0 : P^{(n)} = Q^{(n)} \quad \text{against} \quad \mathcal{H}_1 : P^{(n)} \neq Q^{(n)}. \tag{1}$$

Our theoretical results in subsequent sections will often be in the asymptotic case as $n \to \infty$. For this, we assume that there are two sequences of models $\left(P^{(n)}\right)_{n \geq 1}$ and $\left(Q^{(n)}\right)_{n \geq 1}$, and the sequences are identical under the null hypothesis $\mathcal{H}_0$. We derive asymptotic powers of the proposed tests assuming certain separation rates under the alternative hypothesis.

## 3 Testing large population of graphs $(m \to \infty)$

Before proceeding to the case of small population size, we discuss a baseline approach that is designed for the large $m$ regime ($m \to \infty$). The following discussion provides a $\chi^2$-type test statistic for networks, which is a simplification of Ginestet et al. (2017) under the IER assumption. Given the adjacency matrices $A_{G_1}, \ldots, A_{G_m}$ and $A_{H_1}, \ldots, A_{H_m}$, consider the test statistic

$$T_{\chi^2} = \sum_{i<j} \frac{\left((\overline{A}_G)_{ij} - (\overline{A}_H)_{ij}\right)^2}{\frac{1}{m(m-1)} \sum_{k=1}^{m} \left((A_{G_k})_{ij} - (\overline{A}_G)_{ij}\right)^2 + \frac{1}{m(m-1)} \sum_{k=1}^{m} \left((A_{H_k})_{ij} - (\overline{A}_H)_{ij}\right)^2} , \tag{2}$$

where $(\overline{A}_G)_{ij} = \frac{1}{m} \sum_{k=1}^{m} (A_{G_k})_{ij}$. It is easy to see that under $\mathcal{H}_0$, $T_{\chi^2} \to \chi^2\left(\frac{n(n-1)}{2}\right)$ in distribution as $m \to \infty$ for any fixed $n$. This suggests a $\chi^2$-type test similar to Ginestet et al. (2017). However, like any classical test, no performance guarantee can be given for small $m$ and our numerical results show that such a test is powerless for small $m$ and sparse graphs. Hence, in the rest of the paper, we consider tests that are powerful even for small $m$.

## 4 Testing small populations of large graphs $(m > 1)$

The case of small $m > 1$ for IER graphs was first studied from a theoretical perspective in Ghoshdastidar et al. (2017a), and the authors also show that, under a minimax testing framework, the testing problem is quite different for $m = 1$ and $m > 1$. From a practical perspective, small $m > 1$ is a common situation in neural imaging with only few subjects. The case of $m = 2$ is also interesting for testing between two individuals based on test-retest diffusion MRI data, where two scans are collected from each subject with a separation of multiple weeks (Landman et al., 2011).

Under the assumption of IER models described in Section 2 and given the adjacency matrices $A_{G_1}, \ldots, A_{G_m}$ and $A_{H_1}, \ldots, A_{H_m}$, Ghoshdastidar et al. (2017a) propose test statistics based on

estimates of the distances $\left\|P^{(n)} - Q^{(n)}\right\|_2$ and $\left\|P^{(n)} - Q^{(n)}\right\|_F$ up to certain normalisation factors that account for sparsity of the graphs. They consider the following two test statistics

$$T_{spec} = \frac{\left\|\sum\limits_{k=1}^{m} A_{G_k} - A_{H_k}\right\|_2}{\sqrt{\max\limits_{1 \leq i \leq n} \sum\limits_{j=1}^{n} \sum\limits_{k=1}^{m} (A_{G_k})_{ij} + (A_{H_k})_{ij}}} \;, \text{ and} \tag{3}$$

$$T_{fro} = \frac{\sum\limits_{i<j} \left( \sum\limits_{k \leq m/2} (A_{G_k})_{ij} - (A_{H_k})_{ij} \right) \left( \sum\limits_{k > m/2} (A_{G_k})_{ij} - (A_{H_k})_{ij} \right)}{\sqrt{\sum\limits_{i<j} \left( \sum\limits_{k \leq m/2} (A_{G_k})_{ij} + (A_{H_k})_{ij} \right) \left( \sum\limits_{k > m/2} (A_{G_k})_{ij} + (A_{H_k})_{ij} \right)}} \;. \tag{4}$$

Subsequently, theoretical tests are constructed based on concentration inequalities: one can show that with high probability, the test statistics are smaller than some specified threshold under the null hypothesis, but they exceed the same threshold if the separation between $P^{(n)}$ and $Q^{(n)}$ is large enough. In practice, however, the authors note that the theoretical thresholds are too large to be exceeded for moderate $n$, and recommend estimation of the threshold through bootstrapping. Each bootstrap sample is generated by randomly partitioning the entire population $G_1, \ldots, G_m, H_1, \ldots, H_m$ into two parts, and $T_{spec}$ or $T_{fro}$ are computed based on this random partition. This procedure provides an approximation of the statistic under the null model. We refer to these tests as `Boot-Spectral` and `Boot-Frobenius`, and show their limitations for small $m$ via simulations. Detailed descriptions of these tests are included in Appendix B in the supplementary.

We now propose a test based on the asymptotic behaviour of $T_{fro}$ in (4) as $n \to \infty$. We state the asymptotic behaviour in the following result.

**Theorem 1 (Asymptotic test based on $T_{fro}$).** *In the two-sample framework of Section 2, assume that $P^{(n)}, Q^{(n)}$ have entries bounded away from 1, and satisfy $\max\left\{\left\|P^{(n)}\right\|_F, \left\|Q^{(n)}\right\|_F\right\} = \omega_n(1)$.*

*Under the null hypothesis, $\lim\limits_{n\to\infty} T_{fro}$ is dominated by a standard normal random variable, and hence, for any $\alpha \in (0,1)$,*

$$\mathbb{P}\big(T_{fro} \notin [-t_\alpha, t_\alpha]\big) \leq \alpha + o_n(1), \tag{5}$$

*where $t_\alpha = \Phi^{-1}(1 - \frac{\alpha}{2})$ is the $\frac{\alpha}{2}$ upper quantile of the standard normal distribution.*

*On the other hand, if $\left\|P^{(n)} - Q^{(n)}\right\|_F^2 = \omega_n\left(\frac{1}{m}\max\left\{\left\|P^{(n)}\right\|_F, \left\|Q^{(n)}\right\|_F\right\}\right)$, then*

$$\mathbb{P}\big(T_{fro} \in [-t_\alpha, t_\alpha]\big) = o_n(1). \tag{6}$$

The proof, given in Appendix A, is based on the use of the Berry-Esseen theorem (Berry, 1941). Using Theorem 1, we propose an $\alpha$-level test based on asymptotic normal dominance of $T_{fro}$.

**Proposed Test** `Asymp-Normal`: *Reject the null hypothesis if $|T_{fro}| > t_\alpha$.*

A detailed description of this test is given in Appendix B. The assumption $\left\|P^{(n)}\right\|_F, \left\|Q^{(n)}\right\|_F = \omega_n(1)$ is not restrictive since it is quite similar to assuming that the number of edges is super-linear in $n$, that is, the graphs are not too sparse. We note that unlike the $\chi^2$-test of Section 2, here the asymptotics are for $n \to \infty$ instead of $m \to \infty$, and hence, the behaviour under null hypothesis may not improve for larger $m$. The asymptotic unit power of the `Asymp-Normal` test, as shown in Theorem 1, is proved under a separation condition, which is not surprising since we have access to only a finite number of graphs. The result also shows that for large $m$, smaller separations can be detected by the proposed test.

**Remark 2 (Computational effort).** Note that the computational complexity for computing the test statistics in (3) and (4) is *linear in the total number of edges in the entire population*. However, the bootstrap tests require computation of the test statistic multiple times (equal to number of bootstrap samples $b$; we use $b = 200$ in our experiments). On the other hand, the proposed test compute the statistic once, and is much faster ($\sim$200 times). Moreover, if the graphs are too large to be stored in memory, bootstrapping requires multiple passes over the data, while the proposed test requires only a single pass.

# 5 Testing difference between two large graphs ($m = 1$)

The case of $m = 1$ is perhaps the most interesting from theoretical perspective: the objective is to detect whether two large graphs $G$ and $H$ are identically distributed or not. This finds application in detecting differences in regulatory networks (Zhang et al., 2009) or comparing brain networks of individuals (Tang et al., 2016). Although the concentration based test using $T_{spec}$ is applicable even for $m = 1$ (Ghoshdastidar et al., 2017a), bootstrapping based on label permutation is infeasible for $m = 1$ since there is no scope of permuting labels with unit population size. Tang et al. (2016), however, propose a concentration based test in this case and suggest a bootstrapping based on low rank assumption of the population adjacency. Tang et al. (2016) study the two-sample problem for random dot product graphs, which are IER graphs with low rank population adjacency matrices (ignoring the effect of zero diagonal). This class includes the stochastic block model, where the rank equals the number of communities. Let $G \sim \text{IER}\left(P^{(n)}\right)$ and $H \sim \text{IER}\left(Q^{(n)}\right)$, and assume that $P^{(n)}$ and $Q^{(n)}$ are of rank $r$. One defines the adjacency spectral embedding (ASE) of graph $G$ as $X_G = U_G \Sigma_G^{1/2}$, where $\Sigma_G \in \mathbb{R}^{r \times r}$ is a diagonal matrix containing $r$ largest singular values of $A_G$ and $U_G \in \mathbb{R}^{n \times r}$ is the matrix of corresponding left singular vectors. Tang et al. (2016) propose the test statistic

$$T_{ASE} = \min\left\{\|X_G - X_H W\|_F : W \in \mathbb{R}^{r \times r}, WW^T = I\right\}, \tag{7}$$

where the rank $r$ is assumed to be known. The rotation matrix $W$ aligns the ASE of the two graphs. Tang et al. (2016) theoretically analyse a concentration based test, where the null hypothesis is rejected if $T_{ASE}$ crosses a suitably chosen threshold. In practice, they suggest the following bootstrapping to determine the threshold (Algorithm 1 in Tang et al., 2016). One may approximate $P^{(n)}$ by the estimated population adjacency (EPA) $\widehat{P} = X_G X_G^T$. More random dot product graphs can be simulated from $\widehat{P}$, and a bootstrapped threshold can be obtained by computing $T_{ASE}$ for pairs of graphs generated from $\widehat{P}$. Instead of the $T_{ASE}$ statistic, one may also use a statistic based on EPA as

$$T_{EPA} = \left\|\widehat{P} - \widehat{Q}\right\|_F. \tag{8}$$

This statistic has been used as distance measure in the context of graph clustering (Mukherjee et al., 2017). We refer to the tests based on the statistics in (7) and (8), and the above bootstrapping procedure by `Boot-ASE` and `Boot-EPA` (see Appendix B for detailed descriptions). We find that the latter performs better, but both tests work under the condition that the population adjacency is of low rank, and the rank is precisely known. Our numerical results demonstrate the limitations of these tests when the rank is not correctly known.

Alternatively, we propose a test based on the asymptotic distribution of eigenvalues that is not restricted to graphs with low rank population adjacencies. Given $G \sim \text{IER}\left(P^{(n)}\right)$ and $H \sim \text{IER}\left(Q^{(n)}\right)$, consider the matrix $C \in \mathbb{R}^{n \times n}$ with zero diagonal and for $i \neq j$,

$$C_{ij} = \frac{(A_G)_{ij} - (A_H)_{ij}}{\sqrt{(n-1)\left(P_{ij}^{(n)}\left(1 - P_{ij}^{(n)}\right) + Q_{ij}^{(n)}\left(1 - Q_{ij}^{(n)}\right)\right)}}. \tag{9}$$

We assume that the entries of $P^{(n)}$ and $Q^{(n)}$ are not arbitrarily close to 1, and define $C_{ij} = 0$ when $C_{ij} = \frac{0}{0}$. We show that the extreme eigenvalues of $C$ asymptotically follow the Tracy-Widom law, which characterises the distribution of the largest eigenvalues of matrices with independent standard normal entries (Tracy and Widom, 1996). Subsequently, we show that $\|C\|_2$ is a useful test statistic.

**Theorem 3 (Asymptotic test based on $\|C\|_2$).** *Consider the above setting of two-sample testing, and let $C$ be as defined in (9). Let $\lambda_1(C)$ and $\lambda_n(C)$ be the largest and smallest eigenvalues of $C$.*

*Under the null hypothesis, that is, if $P^{(n)} = Q^{(n)}$ for all $n$, then*

$$n^{2/3}\left(\lambda_1(C) - 2\right) \to TW_1 \quad \text{and} \quad n^{2/3}\left(-\lambda_n(C) - 2\right) \to TW_1$$

*in distribution as $n \to \infty$, where $TW_1$ is the Tracy-Widom law for orthogonal ensembles. Hence,*

$$\mathbb{P}\left(n^{2/3}(\|C\|_2 - 2) > \tau_\alpha\right) \leq \alpha + o_n(1), \tag{10}$$

*for any $\alpha \in (0, 1)$, where $\tau_\alpha$ is the $\frac{\alpha}{2}$ upper quantile of the $TW_1$ distribution.*

*On the other hand, if $P^{(n)}$ and $Q^{(n)}$ are such that $\|\mathbb{E}[C]\|_2 \geq 4 + \omega_n(n^{-2/3})$, then*

$$\mathbb{P}\left(n^{2/3}(\|C\|_2 - 2) \leq \tau_\alpha\right) = o_n(1). \tag{11}$$

The proof, given in Appendix A, relies on results on the spectrum of random matrices (Erdős et al., 2012, Lee and Yin, 2014), and have been previously used for the special case of determining the number of communities in a block model (Bickel and Sarkar, 2016, Lei, 2016). If the graphs are assumed to be block models, then asymptotic power can be proved under more precise conditions on difference in population adjacencies $P^{(n)} - Q^{(n)}$ (see Appendix A.3). From a practical perspective, $C$ cannot be computed since $P^{(n)}$ and $Q^{(n)}$ are unknown. Still, one may approximate them by relying on a weaker version of Szemerédi's regularity lemma, which implies that large graphs can be approximated by stochastic block models with possibly large number of blocks (Lovász, 2012). To this end, we propose to estimate $P^{(n)}$ from $A_G$ as follows. We use a community detection algorithm, such as normalised spectral clustering (Ng et al., 2002), to find $r$ communities in $G$ ($r$ is a parameter for the test). Subsequently $P^{(n)}$ is approximated by a block matrix $\widetilde{P}$ such that if $i, j$ lie in communities $V_1, V_2$ respectively, then $\widetilde{P}_{ij}$ is the mean of the sub-matrix of $A_G$ restricted to $V_1 \times V_2$. Similarly one can also compute $\widetilde{Q}$ from $A_H$. Hence, we propose a Tracy-Widom test statistic as

$$T_{TW} = n^{2/3}\left(\left\|\widetilde{C}\right\|_2 - 2\right), \tag{12}$$

$$\text{where} \quad \widetilde{C}_{ij} = \frac{(A_G)_{ij} - (A_H)_{ij}}{\sqrt{(n-1)\left(\widetilde{P}_{ij}\left(1 - \widetilde{P}_{ij}\right) + \widetilde{Q}_{ij}\left(1 - \widetilde{Q}_{ij}\right)\right)}} \quad \text{for all } i \neq j$$

and the diagonal is zero. The proposed $\alpha$-level test based on $T_{TW}$ and Theorem 3 is the following.

**Proposed Test** `Asymp-TW`: *Reject the null hypothesis if $T_{TW} > \tau_\alpha$.*

A detailed description of the test, as used in our implementations, is given in Appendix B. We note that unlike bootstrap tests based on $T_{ASE}$ or $T_{EPA}$, the proposed test uses the number of communities (or rank) $r$ only for approximation of $P^{(n)}, Q^{(n)}$, and the power of the test is not sensitive to the choice of $r$. In addition, the computational benefit of a distribution based test over bootstrap tests, as noted in Remark 2, is also applicable in this case.

## 6 Numerical results

In this section, we empirically compare the merits and limitations of the tests discussed in the paper. We present our numerical results in three groups: (i) results for random graphs for $m > 1$, (ii) results for random graphs for $m = 1$, and (iii) results for testing real networks. For $m > 1$, we consider four tests. `Boot-Spectral` and `Boot-Frobenius` are the bootstrap tests based on $T_{spec}$ (3) and $T_{fro}$ (4), respectively. `Asymp-Chi2` is the $\chi^2$-type test based on $T_{\chi^2}$ (2), which is suited for the large $m$ setting, and finally, the proposed test `Asymp-Normal` is based on the normal dominance of $T_{fro}$ as $n \to \infty$ as shown in Theorem 1. For $m = 1$, we consider three tests. `Boot-ASE` and `Boot-EPA` are the bootstrap tests based on $T_{ASE}$ (7) and $T_{EPA}$ (8), respectively. `Asymp-TW` is the proposed test based on $T_{TW}$ (12) and Theorem 3. Appendices B and C in the supplementary contain descriptions of all tests and additional numerical results. Matlab codes are provided in the supplementary.[1]

### 6.1 Comparative study on random graphs for $m > 1$

For this study, we generate graphs from stochastic block models with 2 communities as considered in Tang et al. (2016). We define $P^{(n)}$ and $Q^{(n)}$ as follows. The vertex set of size $n$ is partitioned into two communities, each of size $n/2$. In $P^{(n)}$, edges occur independently with probability $p$ within each community, and with probability $q$ between two communities. $Q^{(n)}$ has the same block structure as $P^{(n)}$, but edges occur with probability $(p + \epsilon)$ within each community. Under the null hypothesis $\epsilon = 0$ and hence $Q^{(n)} = P^{(n)}$, whereas under the alternative hypothesis, we set $\epsilon > 0$.

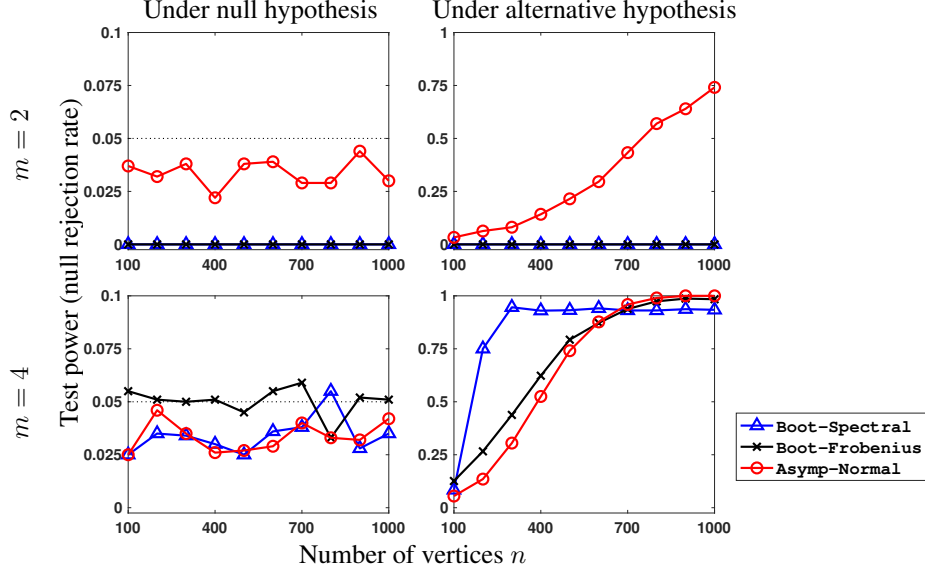

Figure 1: Power of different tests for increasing number of vertices $n$, and for $m = 2, 4$. The dotted line for case of null hypothesis corresponds to the significance level of 5%.

In our first experiment, we study the performance of different tests for varying $m$ and $n$. We let $n$ grow from 100 to 1000 in steps of 100, and set $p = 0.1$ and $q = 0.05$. We set $\epsilon = 0$ and 0.04 for null and alternative hypotheses, respectively. We use two values of population size, $m \in \{2, 4\}$, and fix the significance level at $\alpha = 5\%$. Figure 1 shows the rate of rejecting the null hypothesis (test power) computed from 1000 independent runs of the experiment. Under the null model, the test power should be smaller than $\alpha = 5\%$, whereas under the alternative model, a high test power (close to 1) is desirable. We see that for $m = 2$, only `Asymp-Normal` has power while the bootstrap tests have zero rejection rate. This is not surprising as bootstrapping is impossible for $m = 2$. For $m = 4$, `Boot-Frobenius` has a behaviour similar to `Asymp-Normal` although the latter is computationally much faster. `Boot-Spectral` achieves a higher power for small $n$ but cannot achieve unit power. `Asymp-Chi2` has an erratic behaviour for small $m$, and hence, we study it for larger sample size in Figure 3 (in Appendix C). As is expected, `Asymp-Chi2` has desired performance only for $m \gg n$.

We also study the effect of edge sparsity on the performance of the tests. For this, we consider the above setting, but scale the edge probabilities by a factor of $\rho$, where $\rho = 1$ is exactly same as the above setting while larger $\rho$ corresponds to denser graphs. Figure 4 in the appendix shows the results in this case, where we fix $n = 500$ and vary $\rho \in \{\frac{1}{4}, \frac{1}{2}, 1, 2, 4\}$ and $m \in \{2, 4, 6, 8, 10\}$. We again find that `Asymp-Normal` and `Boot-Frobenius` have similar trends for $m \geq 4$. All tests perform better for dense graphs, but `Boot-Spectral` may be preferred for sparse graphs when $m \geq 6$.

### 6.2 Comparative study on random graphs for $m = 1$

We conduct similar experiments for the case of $m = 1$. Recall that bootstrap tests for $m = 1$ work under the assumption that the population adjacencies are of low rank. This holds in above considered setting of block models, where the rank is 2. We first demonstrate the effect of knowledge of true rank on the test power. We use $r \in \{2, 4\}$ to specify the rank parameter for bootstrap tests, and also as the number of blocks used for community detection step of `Asymp-TW`. Figure 2 shows the power of the tests for the above setting with $\rho = 1$ and growing $n$. We find that when $r = 2$, that is, true rank is known, both bootstrap tests perform well under alternative hypothesis, and outperform `Asymp-TW`, although `Boot-ASE` has a high type-I error rate. However, when an over-estimate of rank is used ($r = 4$), both bootstrap tests break down — `Boot-EPA` always rejects while `Boot-ASE` always accepts — but the performance of `Asymp-TW` is robust to this parameter change.

We also study the effect of sparsity by varying $\rho$ (see Figure 5 in Appendix C). We only consider the case $r = 2$. We find that all tests perform better in dense regime, and the rejection rate of `Asymp-TW` under null is below 5% even for small graphs. However, the performance of both `Boot-ASE` and

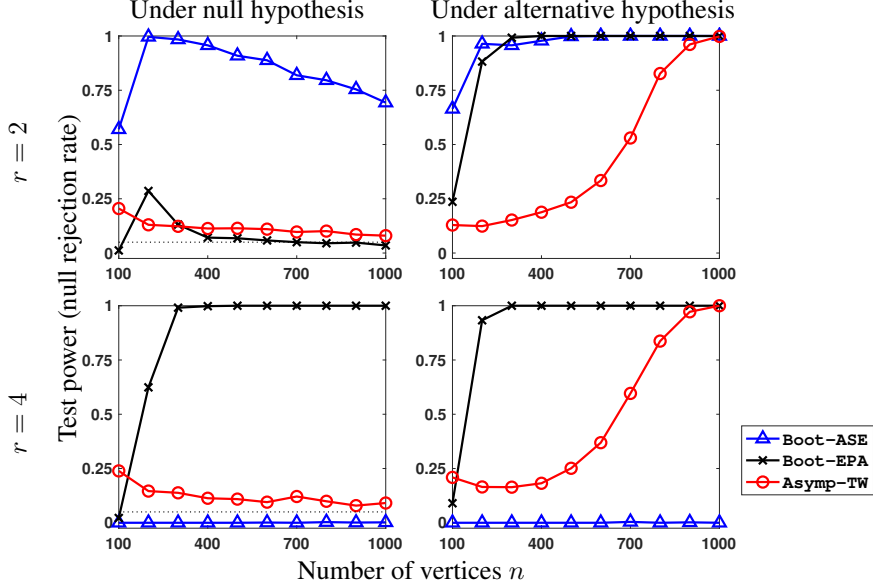

Figure 2: Power of different tests with increase number of vertices $n$, and for rank parameter $r = 2, 4$. The dotted line under null hypothesis corresponds to the significance level of 5%.

`Asymp-TW` are poor if the graphs are too sparse. Hence, `Boot-EPA` may be preferable for sparse graphs, but only if the rank is correctly known.

### 6.3 Qualitative results for testing real networks

We use the proposed asymptotic tests to analyse two real datasets. These experiments demonstrate that the proposed tests are applicable beyond the setting of IER graphs. In the first setup, we consider moderate sized graphs ($n = 178$) constructed by thresholding autocorrelation matrices of EEG recordings (Andrzejak et al., 2001, Dua and Taniskidou, 2017). The network construction is described Appendix C.2. Each group of networks corresponds to either epileptic seizure activity or four other resting states. In Tables 1–4 in Appendix C, we report the test powers and p-values for `Asymp-Normal` and `Asymp-TW`. We find that, except for one pair of resting states, networks for different groups can be distinguished by both tests. Further observations and discussions are also provided in the appendix.

We also study networks corresponding to peering information of autonomous systems, that is, graphs defined on the routers comprising the Internet with the edges representing *who-talks-to-whom* (Leskovec et al., 2005, Leskovec and Krevl, 2014). The information for $n = 11806$ systems was collected once a week for nine consecutive weeks, and two networks are available for each date based on two sets of information ($m = 2$). We run `Asymp-Normal` test for every pair of dates and report the p-values in Table 5 (Appendix C.3). It is interesting to observe that as the interval between two dates increase, the p-values decrease at an exponential rate, that is, the networks differ drastically according to our tests. We also conduct semi-synthetic experiments by randomly perturbing the networks, and study the performance of `Asymp-Normal` and `Asymp-TW` as the perturbations increase (see Figures 6–7). Since the networks are large and sparse, we perform the community detection step of `Asymp-TW` using BigClam (Yang and Leskovec, 2013) instead of spectral clustering. We infer that the limitation of `Asymp-TW` in sparse regime (observed in Figure 5) could possibly be caused by poor performance of standard spectral clustering in sparse regime.

## 7 Concluding remarks

In this work, we consider the two-sample testing problem for undirected unweighted graphs defined on a common vertex set. This problem finds application in various domains, and is often challenging due to unavailability of large number of samples (small $m$). We study the practicality of existing

theoretical tests, and propose two new tests based on asymptotics for large graphs (Thereoms 1 and 3). We perform numerical comparison of various tests, and also provide their Matlab implementations. In the $m > 1$ case, we find that `Boot-Spectral` is effective for $m \geq 6$, but `Asymp-Normal` is recommended for smaller $m$ since it is more reliable and requires less computation. For $m = 1$, we recommend `Asymp-TW` due to robustness to the rank parameter and computational advantage. For large sparse graphs, `Asymp-TW` should be used with a robust community detection step (BigClam).

One can certainly extend some of these tests to more general frameworks of graph testing. For instance, *directed graphs* can be tackled by modifying $T_{fro}$ such that the summation is over all $i, j$ and Theorem 1 would hold even in this case. For *weighted graphs*, Theorem 3 can be used if one modifies $C$ (9) by normalising with variance of $(A_G)_{ij} - (A_H)_{ij}$. Subsequently, these variances can be approximated again through block modelling. For $m > 1$, we believe that *unequal population sizes* can be handled by rescaling the matrices appropriately, but we have not verified this.

### Acknowledgements

This work is supported by the German Research Foundation (Research Unit 1735) and the Institutional Strategy of the University of Tübingen (DFG, ZUK 63).

## Footnotes

[1]Also available at: `https://github.com/gdebarghya/Network-TwoSampleTesting`.

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
