[Supplementary Material]

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

# Appendix for the paper

Here, we provide additional details such as proofs, description of tests, additional numerical results and discussions. Section A provides proofs for the theorems stated in the paper along with a corollary of Theorem 3. Section B provides detailed descriptions of all tests considered in our implementations, both existing tests as well as proposed ones. Section C provides additional numerical results, which we have referred to in the paper.

## A   Proofs for results

In this section, we present the proofs for Theorems 1 and 3, which provide the theoretical foundations for the proposed tests `Asymp-Normal` and `Asymp-TW`, respectively.

### A.1   Proof of Theorem 1

For convenience, we assume $m$ is even. The extension to odd $m$ is straightforward. We also write $P, Q$ instead of $P^{(n)}, Q^{(n)}$ and define

$$\widehat{\mu}_{ij} = \left( \sum_{k \leq m/2} (A_{G_k})_{ij} - (A_{H_k})_{ij} \right) \left( \sum_{k > m/2} (A_{G_k})_{ij} - (A_{H_k})_{ij} \right),$$

$$\widehat{s}_{ij}^2 = \left( \sum_{k \leq m/2} (A_{G_k})_{ij} + (A_{H_k})_{ij} \right) \left( \sum_{k > m/2} (A_{G_k})_{ij} + (A_{H_k})_{ij} \right),$$

$$\widehat{\mu} = \sum_{i<j} \widehat{\mu}_{ij}, \qquad \text{and} \qquad \widehat{s} = \sqrt{\sum_{i<j} \widehat{s}_{ij}^2}.$$

Also let $\mu = \mathbb{E}[\widehat{\mu}] = \frac{m^2}{8}\|P - Q\|_F^2$, $s^2 = \mathbb{E}[\widehat{s}^2] = \frac{m^2}{8}\|P + Q\|_F^2$, and $\sigma^2 = \sum_{i<j} \mathrm{Var}(\widehat{\mu}_{ij})$.

Under the null hypothesis, that is $P = Q$, $\{\widehat{\mu}_{ij} : i < j\}$ are centred mutually independent random variables, and hence, due to the central limit theorem, we can claim that $\frac{\widehat{\mu}}{\sigma}$ converges to a standard normal random variable as $n \to \infty$. The rate of convergence is given by the Berry-Esseen theorem (Berry, 1941) as

$$\sup_x \left| F_{\widehat{\mu}/\sigma}(x) - \Phi(x) \right| \leq \frac{10}{\sigma^3} \sum_{i<j} \mathbb{E}\left[ |\widehat{\mu}_{ij}|^3 \right],$$

where $F_{\widehat{\mu}/\sigma}(\cdot)$ is the distribution function for $\frac{\widehat{\mu}}{\sigma}$. Recall our assumption that the entries are bounded away from 1. Let $\max_{ij} P_{ij} \leq 1 - \delta$ for some $\delta > 0$. Observe that $\widehat{\mu}_{ij}$ is product of two i.i.d. random variables, where each of them is a difference of two binomials. Hence, under $\mathcal{H}_0$, we can compute

$$\sigma^2 = \sum_{i<j} \left( \frac{m}{2} 2P_{ij}(1 - P_{ij}) \right)^2 \geq \frac{m^2\delta^2}{2}\|P\|_F^2,$$

and by using the Cauchy-Schwarz inequality,

$$\mathbb{E}\left[ |\widehat{\mu}_{ij}|^3 \right] \leq \sqrt{\mathbb{E}\left[ \widehat{\mu}_{ij}^2 \right] \mathbb{E}\left[ \widehat{\mu}_{ij}^4 \right]}$$

$$= mP_{ij}(1 - P_{ij}) \left( mP_{ij}(1 - P_{ij})^3 + \frac{m}{2}\left( \frac{m}{2} - 1 \right) 4P_{ij}^2(1 - P_{ij})^2 \right)$$

$$\leq m^2 P_{ij}^2 + m^3 P_{ij}^3 \leq 2m^3 P_{ij}^2.$$

Hence, the Berry-Esseen bound can be written as

$$\sup_x \left| F_{\widehat{\mu}/\sigma}(x) - \Phi(x) \right| \leq 20\sqrt{2}\frac{m^3\|P\|_F^2}{m^3\delta^3\|P\|_F^3} = o_n(1)$$

since $\|P\|_F = \omega_n(1)$. We now compute the probability of type-I error in the following way:

$$\mathbb{P}\big(T_{fro} \notin [-t_\alpha, t_\alpha]\big) = \mathbb{P}\left(\frac{|\widehat{\mu}|}{\widehat{s}} > t_\alpha\right) \leq \mathbb{P}\left(\frac{|\widehat{\mu}|}{\sigma} > (1-\epsilon)t_\alpha\right) + \mathbb{P}\big(\widehat{s}^2 < (1-\epsilon)^2\sigma^2\big) \quad (13)$$

for any $\epsilon \in (0, \frac{1}{2})$. Using the Berry-Esseen bound, we bound the first term as

$$\mathbb{P}\left(\frac{|\widehat{\mu}|}{\sigma} > (1-\epsilon)t_\alpha\right) = 2\big(1 - \Phi((1-\epsilon)t_\alpha)\big) + 2\left|F_{\widehat{\mu}/\sigma}((1-\epsilon)t_\alpha) - \Phi((1-\epsilon)t_\alpha)\right|$$

$$= \alpha + 2\big(\Phi(t_\alpha) - \Phi((1-\epsilon)t_\alpha)\big) + o_n(1)$$

$$\leq \alpha + \epsilon t_\alpha \sqrt{\frac{2}{\pi}} \exp\left(-\frac{t_\alpha^2}{8}\right) + o_n(1)$$

where we use $\epsilon \leq \frac{1}{2}$ in the last step. Taking $\epsilon = \|P\|_F^{-1/2}$ leads to a bound $\alpha + o_n(1)$.

We now deal with the second term in (13). Observe that $\sigma^2 \leq \frac{m^2}{2}\|P\|_F^2 \leq s^2$. Hence, we have

$$\mathbb{P}\big(\widehat{s}^2 < (1-\epsilon)^2\sigma^2\big) \leq \mathbb{P}\big(\widehat{s}^2 < (1-\epsilon)s^2\big)$$

$$= \mathbb{P}\big(s^2 - \widehat{s}^2 > \epsilon s^2\big) \leq \frac{\mathrm{Var}(\widehat{s}^2)}{\epsilon^2 s^4}$$

by the Chebyshev inequality. We can compute the variance term for any $P, Q$ as

$$\mathrm{Var}(\widehat{s}^2) \quad\quad\quad\quad\quad\quad\quad\quad\quad\quad\quad\quad\quad\quad\quad\quad\quad\quad\quad\quad (14)$$

$$= \sum_{i<j} \frac{m^2}{4}\left(P_{ij}(1-P_{ij}) + Q_{ij}(1-Q_{ij})\right)^2 + \frac{m^3}{4}(P_{ij}+Q_{ij})^2\left(P_{ij}(1-P_{ij}) + Q_{ij}(1-Q_{ij})\right)$$

In particular, under $\mathcal{H}_0$, $\mathrm{Var}(\widehat{s}^2) \leq 2m^3\|P\|_F^2$. Using this, the Chebyshev bound is smaller than $\frac{4}{m\epsilon^2\|P\|_F^2} = o_n(1)$ for $\epsilon = \|P\|_F^{-1/2}$. Hence, we obtained the claimed type-I error bound.

For the type-II error rate, we consider the stated separation condition in the form $\frac{m\|P-Q\|_F^2}{\|P+Q\|_F} = \omega_n(1)$. We can bound the error probability as

$$\mathbb{P}\big(T_{fro} \in [-t_\alpha, t_\alpha]\big) \leq \mathbb{P}\left(\frac{|\widehat{\mu}|}{s} \leq 2t_\alpha\right) + \mathbb{P}\big(\widehat{s}^2 \geq 4s^2\big).$$

For the second term, we use the Chebyshev inequality as above to show that the probability is $o_n(1)$ since $\|P+Q\|_F = \omega_n(1)$. For the first term, observe that we have $\frac{\mu}{s} = \omega_n(1)$ under the separation condition, and hence for any fixed $\alpha$, we have $2t_\alpha \leq \frac{\mu}{2s}$ for large enough $n$. So,

$$\mathbb{P}\left(\frac{|\widehat{\mu}|}{s} \leq 2\tau_\alpha\right) \leq \mathbb{P}\left(\frac{\widehat{\mu}}{s} \leq \frac{\mu}{2s}\right) \leq \frac{4\mathrm{Var}(\widehat{\mu})}{\mu^2}.$$

One can compute $\mathrm{Var}(\widehat{\mu})$ similar to (14) to obtain

$$\mathrm{Var}(\widehat{\mu}) \leq \sum_{i<j} \frac{m^2}{4}(P_{ij}+Q_{ij})^2 + \frac{m^3}{4}(P_{ij}-Q_{ij})^2(P_{ij}+Q_{ij})$$

$$\leq \frac{m^2}{8}\|P+Q\|_F^2 + \frac{m^3}{8}\|P-Q\|_F^2\|P+Q\|_F,$$

where the second inequality follows from use of the Cauchy-Schwarz inequality followed by the observation that $\ell_4$-norm is smaller than $\ell_2$-norm. Hence, the error probability is bounded as

$$\mathbb{P}\big(T_{fro} \in [-t_\alpha, t_\alpha]\big) \leq 32\frac{m^2\|P+Q\|_F^2 + m^3\|P-Q\|_F^2\|P+Q\|_F}{m^4\|P-Q\|_F^4} + o_n(1) = o_n(1)$$

under the assumed separation. Hence, the claim.

## A.2 Proof of Theorem 3

We first derive the asymptotic distribution under the null hypothesis. This part is similar to the proof of Lemma A.1 in Lei (2016). Observe that under $\mathcal{H}_0$, $C$ in (9) is a symmetric random matrix, whose entries above the diagonal are independent with mean zero and variance $\frac{1}{n-1}$. Now, let $D$ be a symmetric random matrix with zero diagonal, whose entries above the diagonal are i.i.d. normal with mean zero and variance $\frac{1}{n-1}$. Due to the results of Erdős et al. (2012), we know that $\lambda_1(C)$ and $\lambda_1(D)$ have the same limiting distribution. Lee and Yin (2014) show that $n^{2/3}(\lambda_1(D) - 2) \to TW_1$ as $n \to \infty$, and hence the same conclusion holds for $n^{2/3}(\lambda_1(C) - 2)$. The corresponding result for $-\lambda_n(C)$ can be proved by considering the matrix $-C$. Based on this asymptotic result, we have

$$\mathbb{P}\left(n^{2/3}(\lambda_1(C) - 2) > \tau_\alpha\right) = \frac{\alpha}{2} + o_n(1), \text{ and}$$

$$\mathbb{P}\left(n^{2/3}(-\lambda_n(C) - 2) > \tau_\alpha\right) = \frac{\alpha}{2} + o_n(1),$$

where $\tau_\alpha$ is the $\frac{\alpha}{2}$ upper quantile of the $TW_1$ distribution. Since, $\|C\|_2 = \max\{\lambda_1(C), -\lambda_n(C)\}$, an union bound leads to the stated conclusion under the null hypothesis.

Under the alternative hypothesis, one can see that $\mathbb{E}[C]$ is a re-scaled version of $P - Q$ with each entry being scaled by normalising term of $\sqrt{(n-1)(P_{ij}(1 - P_{ij}) + Q_{ij}(1 - Q_{ij}))}$ (we drop the superscript $n$ for convenience). Under the stated separation condition on $\|\mathbb{E}[C]\|_2$, it is easy to see that $n^{2/3}(\|C\|_2 - 2) \to \infty$ with high probability. So, the probability of the test statistic being smaller than $\tau_\alpha$ is $o_n(1)$. To be precise, we decompose $C$ as $C = \mathbb{E}[C] + (C - \mathbb{E}[C])$, and using Weyl's inequality, we can write

$$\|C\|_2 \geq \|\mathbb{E}[C]\|_2 - \|C - \mathbb{E}[C]\|_2 \geq \|\mathbb{E}[C]\|_2 - \left(2 + n^{-2/3}\tau_\beta\right)$$

with probability at most $\beta + o_n(1)$. The second inequality follows by noting that $(C - \mathbb{E}[C])$ is a mean zero matrix whose spectral norm can be bounded using the arguments stated under the null hypothesis. Hence, $n^{2/3}(\|C\|_2 - 2) \geq n^{2/3}(\|\mathbb{E}[C]\|_2 - 4) - \tau_\beta$ with probability $\beta + o_n(1)$. We set $\tau_\beta = n^{2/3}(\|\mathbb{E}[C]\|_2 - 4) - \tau_\alpha$, and observe that $\tau_\beta = \omega_n(1)$, that is $\beta = o_n(1)$, if $\|\mathbb{E}[C]\|_2 \geq 4 + \omega_n(n^{-2/3})$.

## A.3 Theorem 3 for stochastic block models

We state the following corollary, which provides an understanding of the condition on $\mathbb{E}[C]$ in Theorem 3 under a block model assumption.

**Corollary 4.** *Assume that $P^{(n)}, Q^{(n)}$ correspond to stochastic block models with at most $r_n$ communities, and let $\rho_n = \max_{ij}\left\{P_{ij}^{(n)}, Q_{ij}^{(n)}\right\}$. If $\left\|P^{(n)} - Q^{(n)}\right\|_F^2 = \omega_n\left(nr_n^2\rho_n\right)$, then*

$$\mathbb{P}\left(n^{2/3}(\|C\|_2 - 2) \leq \tau_\alpha\right) = o_n(1). \tag{15}$$

One can observe that if $r_n$ is bounded by a constant and all entries of $P^{(n)}, Q^{(n)}$ are of the same order (same as $\rho_n$), then the above separation condition is similar to the one stated in Theorem 1.

*Proof.* The claim would follow if we show that under the stated separation, the condition on $\mathbb{E}[C]$ used in Theorem 3 holds. In fact, we show that in the present case, $\|\mathbb{E}[C]\|_2 = \omega_n(1)$. For convenience, we simply write $P, Q$ and define $R_{ij} = \sqrt{(n-1)(P_{ij}(1 - P_{ij}) + Q_{ij}(1 - Q_{ij}))} \leq \sqrt{2n\rho_n}$. Note that

$$\mathbb{E}[C_{ij}] = \frac{P_{ij} - Q_{ij}}{R_{ij}},$$

and hence, $\mathbb{E}[C]$ has a block structure with at most $r_n^2$ blocks (ignoring that the diagonal is zero). Thus, there is a diagonal matrix $\Lambda$ such that $\Lambda + \mathbb{E}[C]$ has rank at most $r_n^2$. Note that the diagonal entries of $\Lambda$ are same as the diagonal blocks of $C$, and so, $\|\Lambda\|_2 \leq \max_{ij} \frac{|P_{ij} - Q_{ij}|}{R_{ij}} \leq 2\sqrt{\frac{\rho_n}{(n-1)(1-\rho_n)}} = o_n(1)$

assuming that $\rho_n$ is bounded away from 1. Hence, we can write

$$\|\mathbb{E}[C]\|_2 \geq \|\Lambda + \mathbb{E}[C]\|_2 - \|\Lambda\|_2 \geq \frac{1}{r_n}\|\Lambda + \mathbb{E}[C]\|_F - o_n(1)$$

$$\geq \frac{1}{r_n}\|\mathbb{E}[C]\|_F - o_n(1) \geq \frac{\|P - Q\|_F}{r_n\sqrt{2n\rho_n}} - o_n(1),$$

which is $\omega_n(1)$ under the stated condition. For the second inequality, we use the relation between spectral and Frobenius norms of a matrix with rank $r_n^2$. Finally, Theorem 3 leads to the result. $\quad\square$

## B  Detailed description of tests

In this section, we describe all the tests discussed in this paper. First, we provide description of the asymptotic tests, which include the tests `Asymp-Normal` and `Asymp-TW` proposed in this paper, as well as the large-sample test `Asymp-Chi2`. We next describe the bootstrapped tests `Boot-Spectral` and `Boot-Frobenius`, which are based on approximating the null distribution by randomly permuting the group assignments of the graphs. Tang et al. (2016) provide an algorithmic description of `Boot-ASE`. For completeness, we include this description along with that of `Boot-EPA`, which also generates bootstrap samples based on a low rank approximation of population adjacency. Throughout this section, we refer to the null hypothesis $\mathcal{H}_0$ as the hypotheses that both graphs (or graph populations) have the same population adjacency.

### B.1  Asymptotic tests

We first describe the `Asymp-Normal` test below. In addition to accepting or rejecting the null hypothesis, we also present how to compute the *p-value*, which is defined as the probability that the null hypothesis is true. This is often useful to quantify the amount of dissimilarity between two populations. We use the standard rule of rejecting the null hypothesis when p-value is less than the prescribed significance level $\alpha$. Note that in `Asymp-Normal`, the p-value involves a factor of 2 to take into account both the upper and the lower tail probabilities.

---

**Test `Asymp-Normal`**

**Input:** Graphs $G_1, \ldots, G_m$ and $H_1, \ldots, H_m$ defined on a common vertex set $V$, where $m > 1$; Significance level $\alpha$.
 1: Compute $T_{fro}$ as shown in (4).
 2: p-value $= 2\big(1 - \Phi(-|T_{fro}|)\big)$, where $\Phi$ is the standard normal distribution function.
**Output:** Reject the null hypothesis if p-value $\leq \alpha$.

---

The `Asymp-Chi2` test is listed below. For convenience, we write $T_{\chi^2} = \sum\limits_{i<j} \frac{\tilde{\mu}_{ij}^2}{\tilde{\sigma}_{ij}^2}$, where $\tilde{\mu}_{ij}^2$ and $\tilde{\sigma}_{ij}^2$

denote the numerator and denominator of each term in the summation (2). This notation corresponds to the fact that $\tilde{\mu}_{ij}$ is the sample mean difference for entry $(i,j)$, and $\tilde{\sigma}_{ij}^2$ is an estimate of the variance of $\tilde{\mu}_{ij}$. We note that for sparse graphs and small $m$, the summation in (2) may have terms of the form $\frac{0}{0}$. Hence, we sum only over the set of edges in $\mathcal{C}$ defined below.

---

**Test `Asymp-Chi2`**

**Input:** Graphs $G_1, \ldots, G_m$ and $H_1, \ldots, H_m$, where $m > 1$; Significance level $\alpha$.
 1: Let $\mathcal{C} = \{(i,j) : i < j, \ \tilde{\mu}_{ij} \neq 0 \text{ or } \tilde{\sigma}_{ij} \neq 0\}$, where $\tilde{\mu}_{ij}, \tilde{\sigma}_{ij}$ are defined above.
 2: Compute $T_{\chi^2}$ similar to (2), but sum only over $(i,j) \in \mathcal{C}$.
 3: p-value $= 1 - F_{\chi^2}\left(T_{\chi^2}, \frac{n(n-1)}{2}\right)$, where $F_{\chi^2}(\cdot, \nu)$ is the $\chi^2$-distribution function with degree of freedom $\nu$.
**Output:** Reject the null hypothesis if p-value $\leq \alpha$.

---

We now described `Asymp-TW`, which is the proposed asymptotic test for testing between two given graphs $G$ and $H$ (that is, $m = 1$). As noted in the main paper, this test uses a block model approximation to compute the matrices $\widetilde{P}, \widetilde{Q}$. In the following description, we assume that a partition

of $V$ into $V_1, \ldots, V_r$ is provided as input to the test. For simplicity, we assume that the same partitioning is used for both graphs, but this is not a necessity. In our implementations, we use normalised spectral clustering (Ng et al., 2002) to compute the partition from the average of the two adjacency matrices. A minor difference here is that we use the dominant singular vectors of the normalised adjacency instead of the dominant eigenvectors. This modification is made since the networks could be either homophilic (communities are highly connected) or heterophilic (inter-community links are more frequent as in a bi-partite graph). We also provide an option to externally provide the communities. We use this feature for the real data from Stanford network collection, where we pre-compute the community structure using BigClam (Yang and Leskovec, 2013). From the test statistic $T_{TW}$, we compute the p-value by using available table of distribution function for Tracy-Widom law.[2] The factor of 2 is due to the fact that only the extreme eigenvalues are known to follow the $TW_1$ distribution, and hence, we need union bound for $\|\widetilde{C}\|_2 = \max\left\{\lambda_1(\widetilde{C}), -\lambda_n(\widetilde{C})\right\}$.

---

**Test `Asymp-TW`**

---

**Input:** Graphs $G, H$ defined on vertex set $V$; Partition of $V$ into $V_1, \ldots, V_r$; Significance level $\alpha$.

1: **for all** $V_k$ **do**
2:      **for all** $i, j \in V_k, i \neq j$ **do**
3:          Let $\widetilde{P}_{ij} = \frac{2}{|V_k|(|V_k|-1)} \sum\limits_{i',j' \in V_k : i' < j'} (A_G)_{ij}$ and $\widetilde{Q}_{ij} = \frac{2}{|V_k|(|V_k|-1)} \sum\limits_{i',j' \in V_k : i' < j'} (A_H)_{ij}$.
4:      **end for**
5: **end for**
6: **for all** $V_k, V_l, k \neq l$ **do**
7:      **for all** $i \in V_k, j \in V_l$ **do**
8:          Compute $\widetilde{P}_{ij} = \frac{1}{|V_k||V_l|} \sum\limits_{i' \in V_k, j' \in V_l} (A_G)_{ij}$ and $\widetilde{Q}_{ij} = \frac{1}{|V_k||V_l|} \sum\limits_{i' \in V_k, j' \in V_l} (A_H)_{ij}$.
9:      **end for**
10: **end for**
11: Compute $\widetilde{C}$ and $T_{TW}$ as in (12).
12: p-value $= 2\left(1 - F_{TW_1}(T_{TW})\right)$, where $F_{TW_1}$ is the distribution function for Tracy-Widom law.
**Output:** Reject the null hypothesis if p-value $\leq \alpha$.

---

## B.2 Bootstrap tests

We begin with the description of `Boot-Spectral` and `Boot-Frobenius`. We present both tests together since they follow the same bootstrapping procedure, and only differ in terms of the test statistic. The differences of `Boot-Frobenius` from `Boot-Spectral` are noted in parentheses.

---

**Test `Boot-Spectral` (or `Boot-Frobenius`)**

---

**Input:** Graphs $G_1, \ldots, G_m$ and $H_1, \ldots, H_m$, where $m > 1$; Significance level $\alpha$; Number of bootstraps $b$.

1: Let $T = T_{spec}$ as computed in (3) (or $T = T_{fro}$ in (4)).
2: **for** $i = 1$ **to** $b$ **do**
3:      Randomly split $\{G_1, \ldots, G_m, H_1, \ldots, H_m\}$ into two populations of equal size.
4:      Let $T_i$ be the spectral norm statistic (3) for this split (or Frobenius norm statistic (4)).
5: **end for**
6: p-value $= \frac{1}{b}\left(\left|\{i : T_i \geq T\}\right| + 0.5\right)$, where 0.5 is added for continuity correction.
**Output:** Reject the null hypothesis if p-value $\leq \alpha$.

---

Finally, we present the tests `Boot-ASE` and `Boot-EPA` based on adjacency spectral embedding (ASE) and estimated population adjacency (EPA), respectively. The differences of `Boot-EPA` from `Boot-ASE` are noted in parentheses. Note that these tests compute two approximations of the null distribution — one based on pairs of graphs generated from $\widehat{P}$, and other based on graph pairs generated from $\widehat{Q}$. The p-value is finally computed to ensure that the null is rejected only when the test statistic is in the upper $\alpha$-quantile for both approximate distributions.

**Test** `Boot-ASE` **(or** `Boot-EPA`**)**

---

**Input:** Graphs $G$ and $H$; Significance level $\alpha$; Number of bootstraps $b$.

1: Let $X_G$ and $\widehat{P}$ be the ASE and EPA for graph $G$, respectively (as described in Section 5).
2: Let $X_H$ and $\widehat{Q}$ be the ASE and EPA for graph $H$, respectively.
3: Compute $T = T_{ASE}$ as in (7) (or $T = T_{EPA}$ in (8)).
4: **for** $i = 1$ **to** $b$ **do**
5:     Generate $G_1, G_2 \sim_{\text{iid}} \text{IER}(\widehat{P})$.
6:     Let $T_i$ be the ASE statistic (7) between $G_1, G_2$ (or EPA statistic (8)).
7: **end for**
8: Compute $p = \frac{1}{b}\left(\left|\{i : T_i \geq T\}\right| + 0.5\right)$, where 0.5 is added for continuity correction.
9: **for** $i = 1$ **to** $b$ **do**
10:     Generate $H_1, H_2 \sim_{\text{iid}} \text{IER}(\widehat{Q})$.
11:     Let $T_i'$ be the ASE statistic (7) between $H_1, H_2$ (or EPA statistic (8)).
12: **end for**
13: Compute $p' = \frac{1}{b}\left(\left|\{i : T_i' \geq T\}\right| + 0.5\right)$.
14: p-value $= \max\{p, p'\}$.

**Output:** Reject the null hypothesis if p-value $\leq \alpha$.

---

## C    Additional numerical results and discussions

Here, we provide additional results along with further details for the experiments with real data.

### C.1    Further simulations for random graphs

In this section, we present the figures related to experiments on block models, which we have referred to in the main paper. We have earlier noted that `Asymp-Chi2` has an erratic behaviour for small $m$. This is not surprising since the variance estimates used in (2) are not reliable for small $m$, particularly when the graphs are sparse. We demonstrate this effect even for slightly larger $m$ by comparing `Asymp-Chi2` and `Asymp-Normal` for $m \in \{10, 20, 50, 100, 200\}$. The graph sizes are kept relatively small $n \in \{50, 100, 150, 200\}$. The models are same as the ones used in the experiment of Figure 1.

Figure 3: Power of the asymptotic tests for different values of graph size $n$ and population size $m$. Each row corresponds to a particular test.

Figure 4: Power of different tests for varying levels of sparsity $\rho$ (larger $\rho$ implies denser graphs), and for different values of population size $m$. Each row corresponds to a particular test.

The result, plotted in Figure 3, reveals the undesirable behaviour of `Asymp-Chi2` as the test always has zero rejection rate for $m = 10$. For $m \geq 50$, the test power under alternative hypothesis is 1, but the rejection under null increases with $n$. In particular, rejection rate under null is less than significance level only for $m = 200$ and $n = 50$. Thus, `Asymp-Chi2` is reliable only for $m \gg n$. On the other hand, both Figures 1 and 3 confirm our theoretical observation that the behaviour of `Asymp-Normal` under $\mathcal{H}_0$ does not change with $m$, while its power under $\mathcal{H}_1$ improves for larger $m$.

Figure 4 corresponds to our study related to varying levels of graph sparsity. In this case, the models for $P^{(n)}$ and $Q^{(n)}$ are stochastic block models with same two communities. For $P^{(n)}$, within-class edge probability is $\rho p$ and across-class probability is $\rho q$. We define $Q^{(n)}$ such that the within-class edge probability is $\rho(p + \epsilon)$. Thus, this setting is identical to previous case of Figure 1 for $\rho = 1$. In Figure 4, we fix $n = 500$ and show the rejection rates of the tests for varying sample size $m$ and density $\rho$. The key conclusions are given in the main paper. Additionally, we note the effect of normal dominance in case of `Asymp-Normal`. Recall that $T_{fro}$ does not converge to the normal distribution, but it is dominated by a standard normal random variable. Thus, our threshold for rejection is actually higher than the $\frac{\alpha}{2}$-upper quantile of true asymptotic distribution of $T_{fro}$. This effect is pronounced for dense graphs, where the rejection rate under null is much smaller than the pre-fixed 5% level.

We present a similar study on the effect of sparsity in the case $m = 1$. The results in Figure 5 are based on the above setup, where we have $m = 1$ and vary the the graph size $n$ and the density parameter $\rho$. In this experiment, we use the true rank $r = 2$. This provides an advantage to the bootstrap tests since we observe in Figure 2 that these tests fail when approximation based on a different rank is used. We note that `Boot-ASE` has a high rejection rate under both $\mathcal{H}_0$ and $\mathcal{H}_1$. The rejection rate under $\mathcal{H}_0$ is

Figure 5: Power of different tests with increase number of vertices $n$, and for different levels of sparsity $\rho$. Each row corresponds to a particular test.

smaller for dense graphs, but still above the desired 5% level. For sparse graphs $\rho < 1$, this test is not reliable. On the other hand, `Boot-EPA` performs quite well for both sparse and dense graphs although it uses the same bootstrapping principle. Hence, we may conclude that the test statistic $T_{EPA}$, which was previously not used in the testing literature, is a more useful test statistic. The asymptotic test `Asymp-TW` works well for dense graphs $\rho \geq 1$, but is not reliable in the sparse regime. There can be two potential reasons for this: (i) the approximation of normalisation terms in (9) using $\widetilde{P}$ and $\widetilde{Q}$ is poor in the sparse regime; or (ii) the use of standard spectral clustering for community detection fails in sparse graphs. We believe that the latter reason is more probable since, in a later experiment with sparse real networks, we observe desirable performance from `Asymp-TW` when the community detection is done using BigClam (Yang and Leskovec, 2013).

## C.2 Experiments with EEG recordings of epileptic seizure

In this section, we describe our experiments with networks constructed from EEG recordings of patients with epileptic seizure (Andrzejak et al., 2001). We obtained the data from Dua and Taniskidou (2017), where each EEG recording is divided into several one-second snapshots containing 178 time points ($n = 178$). There are a total of 11500 snapshots available that are classified into five groups:
**Group-1:** Recording of seizure activity;
**Group-2:** Recording of an area with tumour;
**Group-3:** Recording of a healthy brain area;
**Group-4:** Recording of patient with eyes open;
**Group-5:** Recording of patient with eyes closed.

Table 1: Power of `Asymp-Normal` for EEG correlation networks.

|        | G1.1  | G1.2  | G2.1  | G2.2  | G3.1  | G3.2  | G4.1 | G4.2 | G5.1  | G5.2  |
|--------|-------|-------|-------|-------|-------|-------|------|------|-------|-------|
| **G1.1** | 0     | 0.011 | 1     | 1     | 1     | 1     | 1    | 1    | 1     | 1     |
| **G1.2** | 0.011 | 0     | 1     | 1     | 1     | 1     | 1    | 1    | 1     | 1     |
| **G2.1** | 1     | 1     | 0     | 0.003 | 0.009 | 0.008 | 1    | 1    | 1     | 1     |
| **G2.2** | 1     | 1     | 0.003 | 0     | 0.009 | 0.005 | 1    | 1    | 1     | 1     |
| **G3.1** | 1     | 1     | 0.009 | 0.009 | 0     | 0     | 1    | 1    | 1     | 1     |
| **G3.2** | 1     | 1     | 0.008 | 0.005 | 0     | 0     | 1    | 1    | 1     | 1     |
| **G4.1** | 1     | 1     | 1     | 1     | 1     | 1     | 0    | 0    | 1     | 1     |
| **G4.2** | 1     | 1     | 1     | 1     | 1     | 1     | 0    | 0    | 1     | 1     |
| **G5.1** | 1     | 1     | 1     | 1     | 1     | 1     | 1    | 1    | 0     | 0.010 |
| **G5.2** | 1     | 1     | 1     | 1     | 1     | 1     | 1    | 1    | 0.010 | 0     |

In our experiments, we construct networks by thresholding the autocorrelation matrices of the EEG snapshots. The reason for considering such networks is due to their ubiquity in bioinformatics and neuroscience, where most networks are typically derived from correlations or covariances. Moreover, through this setup, we also establish that though the proposed tests are theoretically analysed for edge-independent graphs, they can also be used for other types of networks.

We randomly split each class into four parts of equal size, and compute autocorrelation matrices from the snapshots in each part. Unweighted graphs are obtained by retaining only the largest 10% of correlations (total of 20 graphs). For `Asymp-Normal` test, two graphs are needed for each population. Hence, for each class-$i$, we create two sub-groups $\mathbf{G}i.\mathbf{1}$ and $\mathbf{G}i.\mathbf{2}$, each with two networks. We subsequently test between every pair of the 10 sub-groups — $\mathbf{G}i.\mathbf{1}$ vs. $\mathbf{G}i.\mathbf{2}$ is an instance of null hypothesis while every other pair is an instance of alternative hypothesis. For `Asymp-TW`, we only use the first graph in the sub-group for testing and use $r = 10$ communities for approximation. We run the above setup for 1000 independent trials (the randomness is induced by the splits of the classes during network construction) and report the powers of both tests in Tables 1 and 2.

Table 1 shows that for $\mathbf{G}i.\mathbf{1}$ vs. $\mathbf{G}i.\mathbf{2}$, the null hypothesis is nearly always accepted by `Asymp-Normal` (rejection rate less than 1.1%). In other cases, the rejection is 100% except for $\mathbf{G2}.x$ vs. $\mathbf{G3}.x$ which shows that these two classes have identical behaviour. Table 2 shows that `Asymp-TW` arrives at mostly similar conclusions, but in several cases of alternative hypothesis the power can be much smaller than 1. This is not surprising since the problem is harder for $m = 1$. We note that the authors of the dataset also do not claim that the various rest states can be distinguished, and state that the data is

Table 2: Power of `Asymp-TW` for EEG correlation networks.

|        | G1.1 | G1.2 | G2.1  | G2.2  | G3.1  | G3.2  | G4.1  | G4.2  | G5.1  | G5.2  |
|--------|------|------|-------|-------|-------|-------|-------|-------|-------|-------|
| **G1.1** | 0    | 1    | 1     | 1     | 1     | 1     | 1     | 1     | 1     | 1     |
| **G1.2** | 1    | 0    | 1     | 1     | 1     | 1     | 1     | 1     | 1     | 1     |
| **G2.1** | 1    | 1    | 0     | 0.002 | 0     | 0.001 | 1     | 1     | 0.243 | 0.260 |
| **G2.2** | 1    | 1    | 0.002 | 0     | 0     | 0.001 | 1     | 1     | 0.247 | 0.251 |
| **G3.1** | 1    | 1    | 0     | 0     | 0     | 0     | 1     | 1     | 0.234 | 0.245 |
| **G3.2** | 1    | 1    | 0.001 | 0.001 | 0     | 0     | 1     | 1     | 0.243 | 0.258 |
| **G4.1** | 1    | 1    | 1     | 1     | 1     | 1     | 0     | 0.029 | 0.699 | 0.664 |
| **G4.2** | 1    | 1    | 1     | 1     | 1     | 1     | 0.029 | 0     | 0.647 | 0.667 |
| **G5.1** | 1    | 1    | 0.243 | 0.247 | 0.234 | 0.243 | 0.699 | 0.647 | 0     | 0.049 |
| **G5.2** | 1    | 1    | 0.260 | 0.251 | 0.245 | 0.258 | 0.664 | 0.667 | 0.049 | 0     |

Table 3: Negative logarithm of p-value (averaged over 1000 runs) computed by `Asymp-Normal` for EEG correlation networks.

|  | G1.1 | G1.2 | G2.1 | G2.2 | G3.1 | G3.2 | G4.1 | G4.2 | G5.1 | G5.2 |
|---|---|---|---|---|---|---|---|---|---|---|
| **G1.1** | 0 | 0.7 | 47.3 | 47.5 | 63.6 | 63.5 | 176.2 | 176.1 | 37.6 | 37.5 |
| **G1.2** | 0.7 | 0 | 47.4 | 47.7 | 63.7 | 63.6 | 176.5 | 176.5 | 37.9 | 37.8 |
| **G2.1** | 47.3 | 47.4 | 0 | 0.5 | 1.0 | 1.0 | 331.8 | 332.0 | 37.2 | 37.0 |
| **G2.2** | 47.5 | 47.7 | 0.5 | 0 | 1.0 | 1.0 | 331.6 | 331.9 | 37.1 | 37.1 |
| **G3.1** | 63.6 | 63.7 | 1.0 | 1.0 | 0 | 0.2 | 407.5 | 407.7 | 61.8 | 61.9 |
| **G3.2** | 63.5 | 63.6 | 1.0 | 1.0 | 0.2 | 0 | 407.3 | 407.6 | 61.6 | 62.0 |
| **G4.1** | 176.2 | 176.5 | 331.8 | 331.6 | 407.5 | 407.3 | 0 | 0.3 | 45.7 | 45.3 |
| **G4.2** | 176.1 | 176.5 | 332.0 | 331.9 | 407.7 | 407.6 | 0.3 | 0 | 45.8 | 45.4 |
| **G5.1** | 37.6 | 37.9 | 37.2 | 37.1 | 61.8 | 61.6 | 45.7 | 45.8 | 0 | 0.6 |
| **G5.2** | 37.5 | 37.8 | 37.0 | 37.1 | 61.9 | 62.0 | 45.3 | 45.4 | 0.6 | 0 |

often used for binary setting of Group-1 (seizure) against other rest states. To this end, both tests clearly show that Group-1 is significantly different from all other groups (100% rejection).

A surprising observation from Table 2 ($m = 1$) is that the rejection rate is 100% within Group-1 (**G1.1** vs. **G1.2**), whereas this is not the case for Table 1 ($m = 2$). This agrees with the conclusion of Ghoshdastidar et al. (2017a) that the graph testing problem is fundamentally different for $m = 1$ and $m > 1$. Our intuition is that the networks for seizure activity are significantly different from each other, and hence, rejected by `Asymp-TW`. When we group them ($m > 1$), the fundamental question is whether two groups are identically distributed or not, and hence, the variance within each group is also taken into account. Hence, `Asymp-Normal` detects that **G1.1** and **G1.2** are identical when both graphs in each group are considered.

Although Tables 1 and 2 show that the different groups are typically rejected, they do not clearly show the degree of dissimilarity between two groups. The dissimilarity can quantified in terms of the p-value obtained from the tests. While p-value $\leq 5\%$ leads to rejection, we find that in many cases, the p-value is exponentially small. In Tables 3 and 4, we show the negative logarithm of p-value, that is $-\ln(\text{p-value})$, obtained from `Asymp-Normal` and `Asymp-TW`, respectively. The reported value is the average over 1000 independent runs. We note that the 5% significance level corresponds to $-\ln(\text{p-value}) \approx 3$, and hence, values larger than 3 correspond to rejection. Table 3 shows that this quantity can be as high as 400, and in particular, it shows that Group-4 is most dissimilar from other groups. The results of Table 4 are less conclusive since the maximum reported dissimilarity is only

Table 4: Negative logarithm of p-value (averaged over 1000 runs) computed by `Asymp-TW` for EEG correlation networks.

|  | G1.1 | G1.2 | G2.1 | G2.2 | G3.1 | G3.2 | G4.1 | G4.2 | G5.1 | G5.2 |
|---|---|---|---|---|---|---|---|---|---|---|
| **G1.1** | 0 | 7.727 | 7.727 | 7.727 | 7.727 | 7.727 | 7.727 | 7.727 | 7.727 | 7.727 |
| **G1.2** | 7.727 | 0 | 7.727 | 7.727 | 7.727 | 7.727 | 7.727 | 7.727 | 7.727 | 7.727 |
| **G2.1** | 7.727 | 7.727 | 0 | 0.017 | 0.003 | 0.008 | 7.727 | 7.727 | 1.791 | 1.924 |
| **G2.2** | 7.727 | 7.727 | 0.017 | 0 | 0 | 0.009 | 7.727 | 7.727 | 1.841 | 1.928 |
| **G3.1** | 7.727 | 7.727 | 0.003 | 0 | 0 | 0 | 7.727 | 7.727 | 1.718 | 1.780 |
| **G3.2** | 7.727 | 7.727 | 0.008 | 0.009 | 0 | 0 | 7.727 | 7.727 | 1.823 | 1.889 |
| **G4.1** | 7.727 | 7.727 | 7.727 | 7.727 | 7.727 | 7.727 | 0 | 0.195 | 5.149 | 4.950 |
| **G4.2** | 7.727 | 7.727 | 7.727 | 7.727 | 7.727 | 7.727 | 0.195 | 0 | 4.821 | 4.952 |
| **G5.1** | 7.727 | 7.727 | 1.791 | 1.841 | 1.718 | 1.823 | 5.149 | 4.821 | 0 | 0.366 |
| **G5.2** | 7.727 | 7.727 | 1.924 | 1.928 | 1.780 | 1.889 | 4.950 | 4.952 | 0.366 | 0 |

7.727. This is caused by our use of a pre-computed table for the Tracy-Widom distribution that does not return values arbitrarily close to 1 (see Appendix B for a discussion provided along with the description of the test). However, Table 4 still shows that the networks in Group-5 are relatively less different from those in Groups-2, 3 and 4.

### C.3  Experiments with autonomous systems peering networks

Our second experiment with real networks is based on a collection of networks obtained from the Stanford large network collection (Leskovec and Krevl, 2014). The networks are defined on the set of autonomous systems, which is the technical term for groups of routers that comprise the Internet. The edges correspond to communication between two autonomous systems. The first set of networks, called Oregon-1, are created from data collected by *Oregon route-views* between March 31, 2001 and May 26, 2001 once per week. This set contains 9 networks, one date per week. The second set of networks, called Oregon-2, are based on data collected on the same dates, but the peering information is inferred from a combination of *Oregon route-views, Looking glass, and Routing registry*.

All the networks are defined on a set of $n = 11806$ distinct vertices (autonomous systems), but none of the networks include all vertices, that is, every graph has few isolated vertices. The networks are also quite sparse with the number of edges varying between 22000 to 33000. We view the network collection from the following perspective. For each date, we observe two networks (one from each set) that can be considered as a population of size 2 ($m = 2$). Different dates correspond to different models for the networks, and we test for the similarity across different classes. To this end, we perform `Asymp-Normal` to detect differences, and report $-\ln(\text{p-value})$ for every test in Table 5. It is not surprising to find that the test rejects the null hypothesis at 5% significance for every pair of dates, that is, $-\ln(\text{p-value}) > 3$. The interesting observation is that $-\ln(\text{p-value})$ monotonically increases as the interval between two dates becomes larger, that is, the networks vary significantly over time. This observation is also in conjunction with the findings of Leskovec et al. (2005), where a more qualitative analysis was made based on number of edges and average node degree. We do not report corresponding results for `Asymp-TW` since our current implementation can provide a maximum $-\ln(\text{p-value})$ of at most 7.727, and hence, does not provide any additional information.

We next perform semi-synthetic experiments with Oregon network dataset. We first consider the case of $m = 2$, where we use `Asymp-Normal`. For every pair of networks, we randomly select $k = 118$ vertices (1% of vertex set), and replace the sub-graph by an Erdős-Rényi (ER) graph with edge probability $p$. On Figure 6 (left panel), we show how $-\ln(\text{p-value})$ varies as the edge density of the ER graph increases from $p = 0.2$ to $0.4$, where each line corresponds to one date (one pair of networks) and the results are averaged over 100 runs. We find that $-\ln(\text{p-value})$ increases linearly with $p$, that is, p-value decreases exponentially. The trend is almost similar for every network pair. We also study the effect of adding sparse ER graphs in Figure 6 (right panel). Here we plant an ER graph on a random subset of $k$ vertices, where $k$ varies from 1% to 2% of total number of vertices. However, the planted ER graphs are sparse with $p = \frac{20}{k}$, that is they have constant average degree. We observe a slightly super-linear increase of $-\ln(\text{p-value})$ in this case.

Table 5: Negative logarithm of p-value obtained by `Asymp-TW` for every pair of dates in the Oregon network dataset.

|        | Mar 31 | Apr 7 | Apr 14 | Apr 21 | Apr 28 | May 5 | May 12 | May 19 | May 26 |
|--------|--------|-------|--------|--------|--------|-------|--------|--------|--------|
| Mar 31 | 0      | 13.7  | 25.0   | 36.4   | 59.6   | 77.4  | 96.8   | 106.2  | 135.0  |
| Apr 7  | 13.7   | 0     | 6.5    | 15.2   | 31.0   | 45.7  | 61.1   | 69.7   | 93.4   |
| Apr 14 | 25.0   | 6.5   | 0      | 6.0    | 17.9   | 29.6  | 42.5   | 50.2   | 71.4   |
| Apr 21 | 36.4   | 15.2  | 6.0    | 0      | 8.5    | 17.2  | 27.6   | 34.9   | 54.7   |
| Apr 28 | 59.6   | 31.0  | 17.9   | 8.5    | 0      | 5.3   | 12.8   | 22.6   | 45.7   |
| May 5  | 77.4   | 45.7  | 29.6   | 17.2   | 5.3    | 0     | 4.8    | 13.0   | 31.2   |
| May 12 | 96.8   | 61.1  | 42.5   | 27.6   | 12.8   | 4.8   | 0      | 4.7    | 18.3   |
| May 19 | 106.2  | 69.7  | 50.2   | 34.9   | 22.6   | 13.0  | 4.7    | 0      | 5.6    |
| May 26 | 135.1  | 93.4  | 71.4   | 54.7   | 45.7   | 31.2  | 18.3   | 5.6    | 0      |

Figure 6: Variation of $-\ln(\text{p-value})$ for `Asymp-Normal` when Erdős-Rényi subgraphs are planted into the network. Each line corresponds to one of the 9 pairs. The dotted line corresponds to 5% significance level. **(Left)** Subgraph size is 1% of network size, and edge probability is varied. **(Right)** Subgraph size is varied from 1-2% of network size, and edge probability is decreased.

Finally, we consider a semi-synthetic experiment with $m = 1$, where we use `Asymp-TW`. For each of the 18 networks, we randomly select #e pairs of vertices and toggle their connection, that is, if an edge is present then we remove it, or the reverse. We vary #e from 0 to 300 in steps of 25. Figure 7 reports the values for $-\ln(\text{p-value})$ (averaged over 100 runs) for each network. We present the results in two panels corresponding to the two datasets Oregon-1 and Oregon-2. Surprisingly, we find that $-\ln(\text{p-value})$ rapidly increases with #e although the number of perturbed edges are much smaller than the total of $\binom{11806}{2}$ possible pairs. We also find that the networks in each collection have a similar trend, and the Oregon-2 networks show a slightly smaller value than Oregon-1. This is possibly because the Oregon-2 networks are more dense than their Oregon-1 counterparts.

We conclude our discussion with some implementation details for `Asymp-TW` in this setup related to the community detection step. Since the networks are large and sparse, standard spectral clustering fails to return reasonable communities. Hence, we use BigClam (Yang and Leskovec, 2013), which is suitable for finding a large number of communities in a large network. The method returns multiple community assignments for some vertices and does not make any assignments for few. We use BigClam to find an initial set of 50 overlapping communities from the union of all graphs, and then resolve cases of overlap or no-assignments by assigning vertices to communities with which they have maximum connection. These pre-computed communities are used for the purpose of approximation in `Asymp-TW` test. The above results in Figure 7 show that the use of `Asymp-TW` in conjunction BigClam provides reliable results, and hence, we believe that `Asymp-TW` is applicable even in the sparse regime provided that it is used with a reasonable community detection algorithm.

Figure 7: Variation of $-\ln(\text{p-value})$ for `Asymp-TW` when a random set of #e out of $\binom{n}{2}$ edges are inserted/deleted. The dotted line corresponds to 5% significance level. **(Left)** Each line corresponds to one of the 9 networks from Oregon-1 set. **(Right)** Each line is for a network from Oregon-2 set.