[Reviews · NeurIPS 2018]

Reviewer 1



This paper studies the problem of two-sample testing of large graphs under the inhomogeneous Erdos Renyi model. This model is pretty generic, and assumes that an undirected edge (ij) is in the graph with probability P_{ij} independently of all other edges. Most generically the parameter matrix P could be anything symmetric (zero diagonal), but common models are stochastic block model or mixed membership stochastic block model, which both result in P being low rank. Suppose there were two random graph distributions, parameterized by matrices P and Q, and the goal is to test whether P = Q or not (the null hypothesis being that they are equal). They assume that the graphs are vertex-aligned, which helps as it reduces the problem of searching over permutations to align the graphs. If one has many samples from each distribution, and the number of samples goes to infinity, then there are kernel based tests constructed based on comparing the graph adjacency or Laplacian matrices. This paper focuses on the setting where there are only a few samples from each distribution, or even possibly only one sample from each distribution. When the number of samples per distribution is greater than 1, then the paper uses test statistics previously introduced by Ghoshdastidar, one which looks at the spectral norm of differences between the mean adjacency matrix of the two distributions, and the other which looks at correlation of edges in the average adjacency matrices of the two distributions. The limitation of previous tests was that the theoretical thresholds used in the test are too large (for moderate graph sizes), so they estimate threshold by bootstrapping, which can be quite expensive. The contribution of this paper is to provide an asymptotic analysis of these test statistics as the number of vertices goes to infinity, showing that the test statistic is asymptotically dominated by a standard normal random variable. This suggests an easy to compute threshold to use for the hypothesis test which does not require bootstrapping samples to compute. The asymptotics are as the number of vertices goes to infinity rather than the number of samples going to infinity, but the separation condition between P and Q that the test can distinguish between decreases as the number of samples per distribution increases. When there is only one sample per distribution, most previous methods are limited to assuming that P and Q are low rank. The authors propose a test based on asymptotic distribution of eigenvalues of a matrix related to the difference between adjacency matrices (but each entry scaled by a term related to the variances of that edge). Basically they show that if one knew the parameter matrices P and Q, then the extreme eigenvalues of matrix C defined in eq (9) follow the Tracy-Widom law, thus the spectral radius of C is a useful test statistic. Since in practice P and Q are not known, the authors then propose to estimate P and Q using community detection algorithms such as spectral clustering, where P and Q are approximated by stochastic block models. I think this last part is more heuristic as the authors did not mention whether the theoretical result in spectral clustering are tight enough to provide analogous results as Theorem 4 in the setting where P and Q are estimated. I enjoyed reading the paper and found the results quite interesting and clearly presented. Comment about organization: I feel that since the formal problem statement pertains to the entire paper (m > 1 and m=1), it shouldn't be a subsection within the m \to \infty section; maybe make the problem statement its own section.

Reviewer 2



Summary: This paper seeks to develop a computationally inexpensive method for a two-hypothesis test of networks. Two tests are proposed, one for a large number of network replicates and one for a small number. In these tests, theoretical asymptotic distributions for the test statistic are derived, providing an alternative to the myriad bootstrapped network comparison tests out there. The referee recommends a weak acceptance. The derivation of asymptotic distributions is useful, although there are plenty of seemingly adequate comparison tests for networks of fixed size (see reference below). The author does not adequately motivate his particular test. For example, the IER model assumption, particularly given the motivation of brain networks, seems dubious. Nonetheless, the results are theoretically interesting. -Hypothesis testing for networks with theoretically known H0-distributions for the test statistic have been developed: Minh Tang, Avanti Athreya, Daniel L. Sussman, Vince Lyzinski, and Carey E. Priebe. A nonparametric two sample hypothesis testing problem for random dot product graphs. E-print, arxiv.org, 2014. URL http: //arxiv.org/abs/ 1409.2344. -What about the problem of networks varying in size? The problem of comparing networks of fixed size, and even of fixed vertex set, seems well worn. Spectral norms have known theoretical distributions and are also computationally cheap. - What is the Tracy-Widom law? As this is used in the statement of a main result, the author needs to at least sketch this.

Reviewer 3



Clarity - the paper is very well written and clear Originality - the paper is original; the results of paper extend the ongoing (and exciting) work of statistically testing differences between networks. Quality - strong theoretical paper. The underlying assumption of the IER model is a bit simplistic as all edges are assumed conditionally independent. That being the case, the tests arguably do not capture differences in say the number of triangles or the clustering present in two networks. I believe the work is useful, but certainly more needs to be done in the context of more realistic graph structures. Significance - the problem of testing differences in networks is an important and challenging problem. I agree with the authors that this is particularly apparent in the applications like EEG and fMRI when one wants to decide whether the network models of the brain of two individuals are statistically different. I think that this work is useful and provides some intuition as to how to test differences between more sophisticated graph models; however, overall I think that the analysis of the IER model is a bit simplistic.